Genomic profiling and expression analysis of the diacylglycerol kinase gene family in heterologous hexaploid wheat

Jia Xiaowei 1
Si Xuyang 1
Jia Yangyang 1
Zhang Hongyan 1
Tian Shijun 1
Li Wenjing 1
Zhang Ke zhangke0126@163.com 2
Pan Yanyun pyycell@163.com 1
1 College of Life Science, Hebei Agricultural University/Key Laboratory of Hebei Province for Plant Physiology and Molecular Pathology , Baoding , Hebei , China
2 College of Agronomy, Hebei Agricultural University/State Key Laboratory of North China Crop Improvement and Regulation/Key Laboratory of Crop Growth Regulation of Hebei Province , Baoding , Hebei , China
Arora Gunjan
Electronic publication date: 2021 Dec 14
Publication date: 2021
Volume: 9
Electronic Location ID: e12480
Received 2020 Dec 16; Accepted 2021 Oct 21
Copyright: ©2021 Jia et al.
Copyright year: 2021
Copyright holder: Jia et al.
License: This is an open access article distributed under the terms of the Creative Commons Attribution License, which permits unrestricted use, distribution, reproduction and adaptation in any medium and for any purpose provided that it is properly attributed. For attribution, the original author(s), title, publication source (PeerJ) and either DOI or URL of the article must be cited.
License URL: https://creativecommons.org/licenses/by/4.0/

Keywords: Abiotic stress, Wheat, Diacylglycerol kinase, Gene family

Funding: The National Science Foundation of China (NSFC) Project 31900623 The Natural Science Foundation of Hebei Province in China C2017204095 C2019204041 The Science and Technology Research Project of University of Hebei Province, China ZD2017039 QN2020237 The purchase of instruments, reagents for research, and the fee for English editing of manuscript C2019204041 This work was supported by grants from the National Science Foundation of China (NSFC) (Project 31900623), the Natural Science Foundation of Hebei Province in China (Projects C2017204095 and C2019204041); the Science and Technology Research Project of University of Hebei Province, China (ZD2017039 and QN2020237). Each of the funders provided financial support for the purchase of instruments, reagents for research, and the fee for English editing of manuscript was offered by C2019204041. The funders had no role in study design, data collection and analysis, decision to publish, or preparation of the manuscript.

==============================
The inositol phospholipid signaling system mediates plant growth, development, and responses to adverse conditions. Diacylglycerol kinase (DGK) is one of the key enzymes in the phosphoinositide-cycle (PI-cycle), which catalyzes the phosphorylation of diacylglycerol (DAG) to form phosphatidic acid (PA). To date, comprehensive genomic and functional analyses of DGKs have not been reported in wheat. In this study, 24 DGK gene family members from the wheat genome (TaDGKs) were identified and analyzed. Each putative protein was found to consist of a DGK catalytic domain and an accessory domain. The analyses of phylogenetic and gene structure analyses revealed that each TaDGK gene could be grouped into clusters I, II, or III. In each phylogenetic subgroup, the TaDGKs demonstrated high conservation of functional domains, for example, of gene structure and amino acid sequences. Four coding sequences were then cloned from Chinese Spring wheat. Expression analysis of these four genes revealed that each had a unique spatial and developmental expression pattern, indicating their functional diversification across wheat growth and development processes. Additionally, TaDGKs were also prominently up-regulated under salt and drought stresses, suggesting their possible roles in dealing with adverse environmental conditions. Further cis-regulatory elements analysis elucidated transcriptional regulation and potential biological functions. These results provide valuable information for understanding the putative functions of DGKs in wheat and support deeper functional analysis of this pivotal gene family. The 24 TaDGKs identified and analyzed in this study provide a strong foundation for further exploration of the biological function and regulatory mechanisms of TaDGKs in response to environmental stimuli.

Introduction

As the major components of biomembrane and the basis for lipid signaling, lipids have been proposed to be among the most important biomolecules found in plant tissues. The inositol phospholipid signaling system is based on the metabolism of phosphoinositide (PI), which is synthesized principally through the PI-cycle [PI → phosphatidylinositol 4-bisphosphate (PI4P) → phosphatidylinositol 4,5-bisphosphate [PI (4, 5) P2] → diacylglycerol (DAG) → phosphatidic acid (PA) → cytidine diphosphate-diacylglycerol (CDP-DAG) → PI] (Hou, Ufer & Bartels, 2016; Heilmann & Heilmann, 2015). These derivatives and catalytic enzymes, such as phosphoinositide-specific phospholipases C (PI-PLCs, PLCs) and diacylglycerol kinase (DGK), play pivotal roles in the production of the lipid messengers that mediate plant growth, development, and responses to biotic and abiotic cues (Wang et al., 2014; Kimura, Jennings & Epand, 2016). Steps in the PI-cycle include conversion of DAG to PA catalyzed by DGK. All DGKs have a conserved catalytic domain (DGKc) and a diacylglycerol kinase accessory (DGKa) domain, and some harbor a C1 domain (Hou, Ufer & Bartels, 2016; Wang et al., 2014; Arisz, Testerink & Munnik, 2009).

Plant DGKs have been cloned from many plant species, including Arabidopsis thaliana (Katagiri, Mizoguchi & Shinozaki, 1996; Gómez-Merino et al., 2005), tomato (Snedden & Blumwald, 2000), maize (Zea mays) (Sui et al., 2008), rice (Oryza sativa) (Ge et al., 2012), Malus prunifolia Li et al., 2015a; Li et al., 2015b, tobacco (Cacas et al., 2017), and soybean (Glycine max) (Carther et al., 2019). Based on their gene architectures, evolutionary relationships, and sequences, plants DGKs have been classified into three distinct phylogenetic clusters (Arisz, Testerink & Munnik, 2009). In the genome of Arabidopsis thaliana, seven AtDGK genes are present, of which AtDGK1/2 fall into cluster I, AtDGK3/4/7 into cluster II, and AtDGK5/6 into cluster III (Arisz, Testerink & Munnik, 2009; Katagiri, Mizoguchi & Shinozaki, 1996). AtDGK2/4/7 were expressed in E. coli, respectively, and the recombinant proteins demonstrated kinase activity in vitro (Gómez-Merino et al., 2005; Gómez-Merino et al., 2004; Mulaudzi et al., 2011). The expression patterns of various AtDGK genes reflect their different physiological roles in plant growth, development, and responses to stimuli. Multiple members of the DGK family responded to salt or drought in many plant species, including maize (Sui et al., 2008), apple Li et al., 2015a; Li et al., 2015b, and soybean (Carther et al., 2019). PA has been proposed to be a pivotal second messenger in plants, and its synthesis has been reported to be induced in response to ethylene (Munnik & Musgrave, 2001; Testerink et al., 2007), abscisic acid (Zhang et al., 2004), wounding and Nod factors (Munnik & Musgrave, 2001), osmotic pressure (Munnik et al., 2000; Testerink et al., 2004), cold (Ruelland et al., 2002), salinity (McLoughlin et al., 2013), temperature changes (Arisz et al., 2013), pathogens (Zhang & Xiao, 2015; De Jong et al., 2004), and drought (Li et al., 2015a; Li et al., 2015b). AtDGK1 and 2 were expressed in the roots and leaves and shown to play a role in responding to cold stress (Katagiri, Mizoguchi & Shinozaki, 1996; Gómez-Merino et al., 2004). AtDGK4 is highly expressed in the pollen and appears to regulate pollen tube growth by binding to nitric oxide (Mulaudzi et al., 2011). In Arabidopsis, AtDGK7 is expressed ubiquitously, but especially in flowers and young seedlings (Gómez-Merino et al., 2005). OsDGK1 is intensively expressed in roots system, and modulates the growth and development of roots, which are associated with lipid mediators that control rice root architecture (Yuan et al., 2019).

Other plant DGK homologs have also been deposited into GenBank, including those in hybrid Populus (BU828590), grape (CB981130), apricot Prunus armeniaca, (CB821694), and wheat (BT009326) (Gómez-Merino et al., 2004). Wheat (Triticum aestivum L.) is an important crop that is limited in terms of both productivity and quality by drought, salinity and low temperatures (Khan et al., 2019; Miransari & Smith, 2019; Winfield et al., 2010; Tricker et al., 2018). Several studies have shown that the phosphoinositide signaling pathway plays a pivotal role at various developmental stages and in abiotic stress responses in common wheat (Munnik & Testerink, 2009; Munnik & Vermeer, 2010). As early as 1992, the activity of DGK and phospholipase C (PLC), two key members in phosphoinositide signaling, were demonstrated in highly purified plasma membrane isolates from roots of wheat (Lundberg & Sommarin, 1992; Melin et al., 1992; Pical et al., 1992). Additionally, overexpression of the wheat phospholipase D gene TaPLDα has been shown to enhance tolerance to drought and osmotic stress in transgenic Arabidopsis thaliana (Wang et al., 2014). Previously, we had measured the TaPLC expression patterns at both the transcriptional and protein levels in response to drought and salinity stress, and our research indicated the possible roles of TaPLC1 in mediating seedling growth and responding to drought and salinity stress (Zhang et al., 2014). However, the molecular functions of TaDGKs, as lipid messengers in wheat remain unclear. As an allohexaploid genome, the wheat genome may contain many homologous genes, and there may be evolutionary differences and functional differentiation among these genes. Thus, it is necessary to identify all TaDGK genes in wheat, and conduct evolutionary and functional analysis to better understand their roles. Whole-genome sequencing has provided key insights into new methods for investigating genes in wheat. In this study, we conducted a genome-wide survey and analysis of TaDGK family members in genome-wide studies of wheat. We also analyzed TaDGKs expression patterns across various tissues, and under drought or salinity stress. Our goal was to improve the current understanding of the involvement of DGK genes involvement in wheat response mechanisms to abiotic stress as a foundation for future utilization of genetic engineering in agricultural production.

Materials and methods

Identification and chromosomal location of TaDGKs

To study the TaDGK-coding genes in common hexaploid wheat, several methods were employed. We first performed a local BLASTN against the wheat genome database (http://plants.ensembl.org/index.html) by using identified DGK sequences from Arabidopsis and rice as queries to find their homologous genes, and the whole-genome data were downloaded from the Ensembl Plants of wheat (ftp://ftp.ensemblgenomes.org/pub/plants/release-51/fasta/triticum_aestivum). Additionally, all sequences that were acquired by a search of the gene name “diacylglycerol kinase”, were submitted to WheatExp (https://wheat.pw.usda.gov/WheatExp/) to confirm the presence of putative DGK genes in wheat. Protein motifs were queried using the NCBI BLASTP program (http://blast.ncbi.nlm.nih.gov/Blast.cgi?PROGRAM=blastpPAGE_TYPE=BlastSearchLINK_LOC=blasthome), and those that lacked the conserved DGK catalytic domain were ignored. Then we searched again with the conserved sequence of the putative DGK genes against the wheat genome database to ensure that all TaDGKs would be obtained. Chromosomal localization of genes was obtained from Ensemblplant. The online ExPASY Molecular Biology Server (http://web.expasy.org/protparam/) and TMHMMServerv.2.0 (http://www.cbs.dtu.dk/services/TMHMM-2.0/) tools were used for sequence analyses.

The genome data of progenitor species was downloaded from Ensembl Plants, including Triticum urartu (ftp://ftp.ensemblgenomes.org/pub/plants/release-51/fasta/aegilops_tauschii), Aegilops tauschii (ftp://ftp.ensemblgenomes.org/pub/plants/release-51/fasta/triticum_urartu/dna/), and Triticum dicoccoides (ftp://ftp.ensemblgenomes.org/pub/plants/release-51/fasta/triticum_dicoccoides).

Multiple sequence alignments, phylogenetic analysis and gene ontology (GO) enrichment

Multiple sequence alignments of DGK sequences were created using DNAMAN and MEGA7.0 programs with default parameters. Again, using MEGA7.0, all DGK sequences acquired from the NCBI protein database, including those from Arabidopsis, rice, soybean, apple, maize, and wheat, were aligned to analyze their evolutionary relationships, again using the default parameters, and a phylogenetics tree was produced using the neighbor-joining (NJ) method with 1,000 replicates to determine bootstrap support. Additionally, to better understand the biological pathways that the DGK genes were involved in, GO enrichment analysis was performed using Gene Ontology Enrichment Analysis software (Altshuler-Keylin et al., 2016). GO data was downloaded from the GO Ontology database (http://www.geneontology.org) Gene Ontology (Gene Gene Ontology Consortium, 2021; Ashburner et al., 2000). The annotation version is DOI: 10.5281/zenodo.4735677, and the release date is 2021-05-01 (Table S1).

Sequence analysis methods

The exon-intron structures and motifs of DGK genes were generated online with Evolview (https://www.evolgenius.info/evolview/#login). Transposable element analyses were conducted with Giri Repbase (https://www.girinst.org/censor/index.php). Protein domains analyses were conducted using SMART (http://smart.embl-heidelberg.de/) and Pfam (http://pfam.sanger.ac.uk/) databases, and IBS software was used to diagram the domain structures.

Promoters (within 1500 bp upstream of the transcription start site) of DGK genes were surveyed to identify putative cis-regulatory elements using the Plant Cis-acting Regulatory DNA Elements (PLACE) (http://www.dna.affrc.go.jp/PLACE/) and PlantCARE databases (http://bioinformatics.psb.ugent.be/webtools/plantcare/html/). The subcellular localization of DGK was predicted using WoLF PSORT (https://www.genscript.com/wolf-psort.html).

Plant materials and stress treatments

Wheat seeds (Chinese Spring) were provided by Professor Yanyun Pan (Key Laboratory of Hebei Province for Plant Physiology and Molecular Pathology) (Zhang et al., 2014). Seeds were briefly surface-sterilized with ethanol (70%), and then they were immersed in bleach solution (30%) for 10 min. After being washed with sterilized water three times, they were germinated and cultured with a hydroponic system under 16-h days at 24 °C in a growth chamber.

For the salinity and drought treatments, 2-week-old wheat seedlings were treated with 200 mM NaCl and 20% PEG6000, respectively (Zhang et al., 2014; Si et al., 2020). Then, those plants along with control plants were sampled at 0, 1/6, 1/2, 1, 2, 6, 12, and 24 h post-treatment. Additionally, at different developmental stages, including the two-node stage, various tissues, including roots, stems, spikes, and leaves, were sampled and rapidly frozen in liquid nitrogen prior to storage at −80 °C.

RNA extraction and gene expression analysis

Total RNA was extracted using UNIQ-10 Trizol reagent (Sangon Biotech, Shanghai, China). Then, gDNA was eliminated, and first-strand cDNA was synthesized with PrimeScript TM RT reagent Kit (TaKaRa, Dalian, China). In triplicate, quantitative real-time PCR was performed using the Bio-Rad Chromo 4 real-time PCR system (Bio-Rad, Hercules, CA, USA) with SYBR Green PCR master mix (TransGen Biotech, Beijing, China) with the following settings: 95 °C for 5 min; 40 cycles of 95 °C for 10 s; and annealing at 60 °C for 30 s. Three biological replicates were conducted using the primers listed in Table S2, and ACTIN was used as the reference gene. The expression levels of each gene at different time stress points were calculated using the 2−ΔΔCt method for abiotic treatments and the 2−ΔCt method for different tissue, and the results were analyzed with SPSS statistics 17.0 (IBM) using the independent samples t-test.

To analyze the expression profiles of TaDGKs among various tissues, organs, and stress treatments, we downloaded the TaDGKs gene expression data from the Triticeae Multi-omics Center (http://wheatomics.sdau.edu.cn/expression/index.html) (Wang et al., 2020). Based on the TPM value obtained, the heatmap was drawn with RStudio software (version R-3.6.3).

Plasmid vector construction and tobacco transformation

To express TaDGK2A-GFP and TaDGK3A-GFP in Nicotiana benthamiana driven by the 35S promoter, the TaDGK2A and TaDGK3A coding sequences without stop codons were cloned into pSUPER1300, a binary vector with a C-terminal fusion with the GFP tag, by cloning it into the SpeI/KpnI and XbaI/KpnI restriction enzyme sites. DNA sequences for all cloning primers used are listed in Table S2. The 35S:TaDGK2A-GFP and 35S:TaDGK3A-GFP vectors were introduced into N. benthamiana leaves by Agrobacterium-mediated transformation. Tobacco transformation was performed as previously described report (Sparkes et al., 2006).

Subcellular localization

The 35S:TaDGK2A-GFP and 35S:TaDGK3A-GFP vectors were delivered into the epidermis of N. benthamiana leaves by Agrobacterium tumefaciens (EHA105). Three days after infection, images were captured under fluorescence with a Leica TCS SP8 confocal microscope (Leica, Wetzlar, Germany). To capture GFP fluorescence, the excitation light wavelength was set to 488 nm and detected between 510 and 550 nm. To capture DAPI signals, the excitation light wavelength was set to 305 nm and detection at 461 nm. The DAPI staining was performed as described by Andrade & Arismendi (2013).

Figure 1 The distribution of DGK gene family members among wheat chromosomes.

Each cluster is shown in a different color.

Results

Identification and chromosomal distribution of the TaDGK family in wheat

To identify the members of the TaDGK family, the DGK sequences from other plants, including Arabidopsis and rice, were used to conduct local BLAST against wheat genome databases. Furthermore, keywords and protein domain searches were also executed. Ultimately, a total of 24 putative wheat TaDGK genes were identified (Table S3). The Gene Ontology (GO) analysis demonstrated these genes were related to protein kinase C-activating G protein-coupled receptor signaling pathway (GO: 0007205), and function including diacylglycerol kinase activity (GO: 0004143) and NAD+ kinase activity (GO: 0003951) (Figs. S1A and S1B). All the 24 genes were distributed along the 18 of 21 wheat chromosomes (Chr) (Fig. 1). In hexaploid wheat, homologous genes coming from the A, B, and D subgenomes respectively, were deemed to be homoalleles of a single ancestral TaDGK gene that arose from a polyploidization event during genome evolution. Every TaDGK gene exhibited its own orthology among three diploid relatives, with the exception of TaDGK7A2, TaDGK6A2, and TaDGK6D2, which might have been lost throughout the course of asymmetric subgenome evolution. To determine when its homoalleles were lost during evolution, we retrieved the genomes of ancestral species of modern wheat, including Triticum urartu (with an AA diploid genome), Aegilops tauschii (DD diploid genome), and Triticum dicoccoides (AABB tetraploid genome). Notably, we found the orthologs of TaDGK7B2/D2, including TRIUR3_04325, AET7Gv20953100 and TRIDC7BG047160, in the A subgenome of Triticum urartu (AA) and Triticum dicoccoides (AABB), which suggests TaDGK7A2 was lost after the allohexaploidy event. However, the orthologs of TaDGK6B2 was only found in B subgenome of Triticum dicoccoides (AABB), named TRIDC6BG067200, suggesting that TaDGK6A2 and TaDGK6D2 were lost before the allohexaploidy event. These types of events, including intrachromosomal serial duplication and gene loss, may have occurred during the evolution process, whereas a gene loss event appears to have occurred among TaDGKs on wheat chromosomes 7A, 6A and 6D. The phylogenetic analyses on TaDGK6B2, TaDGK7B2/D2, and their orthologs from genomes of ancestral species were shown in Fig. S2. Interestingly, no TaDGK genes were identified on chromosome 4A/B/D, and no DGK homologous sequences were retrieved on chromosome 4 of the ancestral species, suggesting that DGK genes on chromosome 4 did not exist or were lost before the formation of the ancestral species (Lawton-Rauh, 2003).

Taking into account their chromosomal locations, 24 TaDGK genes were identified as TaDGK1A/B/D, TaDGK2A/B/D, TaDGK3A/B/D, TaDGK5A/B/D, TaDGK5A2/B2/D2, TaDGK6A/B/D, TaDGK7A/B/D, TaDGK6B2, and TaDGK7B2/D2 respectively (Fig. 1). The ORFs of these genes were 1,467–2,172 bp, encoding polypeptides of 488–723 amino acids, with predicted molecular weights of 54.25–80.33 kD (Table S4). Their theoretical isoelectric point (Pi) values ranged from 5.79 to 9.04 (Table S4). The nucleotide and amino acid sequences of each gene are shown in Table S3. Interestingly, sequence alignment revealed that BT009326, a reported TaDGK gene, and TaDGK2B were identical in sequence (Table S3).

Multiple sequence alignments and sequence characterization of TaDGK genes

To make sure that the conserved sequences in TaDGKs, we performed multiple alignment of the domains of TaDGKs in cluster I in three phases, for the DGKc domain (Fig. 2A) and each of the two C1 domains (Figs. 2B, 2C, respectively). The alignment revealed that all TaDGKs, like AtDGKs, possessed a conserved DGKc domain containing a putative ATP-binding site with a GXGXXG consensus sequence (the red box in Fig. 2A) (Bunting et al., 1996). TaDGKs have the classical generalized structure as seen in other studied plants (Fig. 3). Additionally, the two C1 domains harbor the sequences HX14CX2CX16-22CX2CX4HX2CX7C and HX18CX2CX16CX2CX4HX2CX11C, respectively (Fig. 3) (Li et al., 2015a; Li et al., 2015b; Carther et al., 2019; Gómez-Merino et al., 2004). By sequence alignment, we found the upstream basic region and extCRD-like domain were substantially conserved, with only slight variation: in the basic region, conserved KA residues were replaced by KV in TaDGK1 A/B/D, and the flanking residue V of extCRD-like was replaced by L in TaDGK5A (Fig. 3).

Figure 2 Multiple alignments of the diacylglycerol kinase catalytic (DGKc) domain (A), diacylglycerol/phorbol ester (DAG/PE)-binding domain C1 (B), and DAG/PE-binding domain 2 (C) in wheat and Arabidopsis thaliana.

The putative ATP-binding site localized (consensus GXGXXG) is enclosed by red rectangle.

Figure 3 A detailed view of domains of DGK genes in cluster I.

The domains of DGKs in cluster I are shown with the predicted location and the sequences of conserved C6/H2 cores; the extended cysteine-rich (extCRD)-like domain and the upstream basic regions are also shown.

Phylogenetic analysis of TaDGK genes

A phylogenetic analysis was conducted based on the inferred protein sequences of Triticum aestivum, Glycine max, Arabidopsis thaliana, Oryza sativa, Zea mays, Malus domestica, Brassica rapa, Sorghum bicolor, Cicer arietinum, and Solanum tuberosum DGK genes (Table S5). The obtained unrooted phylogenetic tree confirmed that these DGK genes were grouped into clusters I, II, and III. TaDGKs were distributed as follows: TaDGK1A/B/D, TaDGK5A/B/D, and TaDGK5A2/B2/D2 were found in cluster I; TaDGK6A/B/D, TaDGK6B2, and TaDGK7A/B/D were found in cluster II; TaDGK2A/B/D, TaDGK3A/B/D, and TaDGK7B2/D2 were found in cluster III (Fig. 4). The phylogenetic tree also revealed that the TaDGKs were more aptly classified with DGKs from the monocots rice and maize than those from the dicots Arabidopsis and soybean. Notably, TaDGK2 and TaDGK7B2/D2 formed a clear paralogous pair.

Figure 4 Phylogenetic analyses of DGK genes in Triticum aestivum (Ta), Glycine max (Gm), Arabidopsis thaliana (At), Oryza sativa (Os), Zea mays (Zm), Malus domestica (Md), Brassica rapa (Bra), Sorghum bicolor (Sb), Cicer arietinum (Ca), and Solanum tuberosum (St).

Clusters are indicated by colors. The following genes are shown with the following corresponding symbols: TaDGKs, black circles; GmDGKs, gray circles; AtDGKs, green circles; OsDGKs, blue circles; ZmDGKs, yellow circles; MdDGKs, red circles; BraDGKs, purple circles; SbDGKs, deep red circles; CaDGKs, yellow circles; StDGKs, light blue circles. The numbers beside the branches indicate the bootstrapvalues that support the adjacent nodes. GenBank accession numbers are listed in Table S4.

Structures and protein motifs of TaDGK genes

Structural analysis was performed to obtain some valuable information about duplication events of gene families in the form of phylogenetic relationships. The exon-intron distributions of TaDGK genes were analyzed using Evolview, which showed that genes in the same cluster were highly similar, especially the ones with closer evolutionary relationships. All of the TaDGKs in clusters II and III had 12 exons, while those in cluster I had 7 exons (Figs. S3A, S3B). The exons of genes within the same cluster showed extraordinary conservation in order and size. Among genes in the same cluster, not only homologs across different wheat chromosomes, but also rice DGK genes had very similar exon-intron structures (Fig. S4). This result showed the orthology of DGK genes across different plant species and suggested that TaDGKs have undergone gene duplications throughout their evolution. In addition, the results of phylogenetic analysis results are further supported by the analyses of sequence and structural features of TaDGK genes. TaDGK homologs in the same clade, such as wheat and rice, had similar exon-intron structures (Figs. S3 and S4).

However, we found some introns of TaDGK1D, TaDGK6B2, TaDGK7A/B/D and TaDGK7B2/D2 was much longer than those of the others homologs. Transposable element analyses were conducted on these introns. Interestingly, we found that these long introns contained many different types of transposon elements (Table S6), which suggested the formation of them occurred through the insertion of transposons as potential controlling elements (Kashkush, Feldman & Levy, 2015). The influence of these transposons on the biological function of TaDGKs is worth further investigation.

The identified motifs were predicted to contain some conserved supersecondary structures that form the domains or the tertiary structures of proteins. MEME analysis revealed 15 distinct motifs in the TaDGK family (Fig. S3C). Three copies of each TaDGK member presented the same motif compositions, and genes belonging to the identical cluster had similar motif compositions. Five motifs, namely 1, 2, 3, 4, 5, 8, and 13, were shared among all TaDGKs. Meanwhile, the motifs 6, 11, 12, and 15 were specific protein motifs specific to cluster I, and motif 10 was specific to cluster III. Motif 9 and 14 existed in clusters II and III. All the sequence logos for these motifs are shown in Fig. S5. In addition, motifs 3, 4, 7, and 12 are related to the diacylglycerol kinase catalytic domain (DGKc), and motifs 1, 5, 8, and 13 are related to the accessory domain (DGKa). Further, motifs 6 and 9 are related to the C1 domain in cluster I.

Protein domains of TaDGKs

By utilizing Illustrator for Biological Sequences (IBS), a schematic diagram was developed for the protein domains in all TaDGKs (Fig. 5). This diagram shows that each TaDGK harbored a diacylglycerol kinase catalytic domain (DGKc) (PF00781) and one accessory domain (DGKa) (PF00609), and demonstrates that TaDGK domains have different distributions based on the conservation of the macro protein domains throughout the evolution of all three clusters and. Furthermore, all the TaDGK genes belonging to cluster I contained two C1 domains (PF00130), i.e., a trans-membrane domain and a DAG/phorbol ester (PE)-binding domain.

Figure 5 Functional domain analysis of TaDGKs.

The numbers shown indicate the position of each amino acid in the protein.

Cis-acting elements in the promoter of TaDGKs

Transcription factors regulate the target genes expressed by binding to cis-regulatory elements (Priest, Filichkin & Mockler, 2009). Cis-regulatory promoter elements of genes, which play key roles in regulating tissue-specific and stress-responsive expression of genes, can reveal transcriptional regulation and potential functions of TaDGKs. Using the PLACE and PlantCARE databases, TaDGK promoters, within 1,500 bp upstream of the transcription start site, were analyzed to identify the putative cis-regulatory elements. CAAT-box and TATA-box elements were overrepresented among all 24 TaDGK promoters (Fig. 6). Moreover, we selected some representative components for subsequent investigation of expression. Thus, the cis-acting elements could be classified into several groups according to abiotic stress responsiveness (water, dehydration, and temperature), biotic stress responsiveness (disease and pathogens), responses to plant hormones (ethylene, auxin, abscisic acid [ABA], gibberellic acid [GA], and salicylic acid [SA]), and metabolic processes (GA biosynthesis) (Fig. 6 and Table S7). In addition, almost all TaDGKs contain MYBCORE (water stress), MYB1AT (dehydration-responsive), and ASF1MOTIFCAMV (abiotic and biotic stress) elements (Fig. 6 and Table S7), which suggests that TaDGKs mediate stress responses in wheat.

Figure 6 Putative regulatory cis-elements in the DGK gene promoters of wheat.

The hormone and stress responsive cis-elements are in red and blue, respectively. The relative positions of elements are labeled with capital letters here and are denoted in Table S6.

Among the several hormones explored through in-silico assessment of RNA-seq experiments, the most significant factor affecting TaDGKs expression is ABA (Fig. S6 and Table S8), which is a major hormone that modulates the ability of plants to survive in harsh, changing environments. The transcript levels of TaDGK2, 3, and 6 were severely suppressed by ABA (Fig. S6 and Table S8), although ABA-responsive elements are located in the promoters of TaDGK1A/B/D and TaDGK5A/B/D (Fig. 6, Table S7). Therefore, one hypothesis is that ABA may influence the expression of TaDGK2, 3 and 6 in some unknown indirect way.

Expression profiles of TaDGK in various tissues

We conducted a microarray-based expression pattern analysis of TaDGK genes using publicly available datasets from the wheat gene expression database hosted by the Triticeae Multi-omics Center. All TaDGK genes members were determined to have some level of tissue-specific expression, and none were constitutively expressed in all investigated tissues (Fig. 7A and Table S8). TaDGK6A/B/D, TaDGK2A/B/D, TaDGK5A/B/D, and TaDGK5A2/B2/D2 showed high expression levels in roots. TaDGK7B2/D2 and TaDGK6B2 showed high expression in spikes. TaDGK1A/B/D and TaDGK3A/B/D showed high expression in grain. Almost all TaDGK genes showed low expression in leaves.

Figure 7 Tissue expression analysis of TaDGK genes.

(A) RNA-seq data analysis results. The data for the analysis of gene expression in roots, stems, leaves, spikes, and grains were retrieved from the Triticeae Multi-omics Center. The color scale at the right of the heat map indicates the relative expression levels, where light blue and red indicate low and high expression, respectively. (B) Analysis of TaDGK genes expression in various tissues by real-time PCR with degenerate Primer. (C) Analysis of TaDGK genes expression in various tissues by real-time PCR with specific primer. Actin was used as the reference gene. Mean values were obtained from three replicates. Vertical bars indicate standard deviations.

Real-time PCR was also carried out to investigate the expression patterns of TaDGKs with degenerate or specific primers in various organs. The results revealed that there were significant differences in the expression levels among the members of TaDGK family in wheat. For example, the expressions of TaDGK5 (TaDGK5A/B/D), TaDGK6 (TaDGK6A/B/D) and TaDGK3A were relatively high, while the transcripts of TaDGK5A2 and TaDGK7-2 (TaDGK7/B2/D2) were almost undetectable in wheat (Figs. 7B and 7C). In addition, consistent with the results of microarray data, the transcription levels of each member of TaDGK family vary greatly in different tissues. Most of the TaDGK genes were low expressed in leaves, but high expressed in roots and spikes (Figs. 7B and 7C). The highest expression of TaDGK3A was found in the stem, which contradicts the results of RNA-seq data (Fig. 7C). Interestingly, the expression of some genes, such as TaDGK6B2, TaDGK7A/B/D and TaDGK7B2/D2 are very low in wheat tissues (Fig. 7C). Gene structure analysis showed that all of these genes contained some long introns with transposon insertion, suggesting that the transposons may regulate TaDGKs biological functions by blocking their transcription (Fig. S3 and Table S6). Apparently, the differences in expression patterns also suggest functional differentiation of these genes.

TaDGK expression patterns under salinity and drought stress

The promoters of almost all TaDGKs were enriched for abiotic stress responsive elements, strongly suggesting the potential functions of TaDGK genes in responses to salinity or drought stress. In order to further functional studies, we selected four members of TaDGK family as target genes, which could be amplified to obtain the full length of CDS. Unfortunately, we did not obtain the TaDGKs CDS from Cluster II, possibly due to transposon insertion and high GC content in gene sequence (Tables S3 and S6). Accordingly, we tested the expression patterns of TaDGKs at the transcriptional level and found that expression of TaDGK genes was induced under stress. After only 10 min of salt treatment, the mRNA abundance of four tested TaDGK genes—TaDGK2A/3A/5B/5A2—increased rapidly, with about 2-fold higher expression than controls (at 0 h). Three genes—TaDGK2A/5B/5A2—that were highly expressed in the roots were significantly induced after 12 h, increasing by 25-, 18-, and 22-fold respectively, with subsequent gradual down-regulations of expression (Fig. 8A).

Figure 8 Analysis of TaDGK genes expression under salt and drought stress conditions.

(A) qRT-PCR results of TaDGK under salt stress conditions. Wheat leaves were sampled after 0,1/6, 0.5, 1, 2, 6, 12, and 24 h of treatment with 200 mM NaCl. (B) qRT-PCR results for TaDGK genes under drought conditions. Wheat leaves were sampled after 0, 1/6, 0.5, 1, 2, 6, 12, and 24 h of treatment with 20% PEG. Actin served as the reference gene. Mean values were obtained from three replicates. Vertical bars indicate standard deviations. Asterisks above error bars indicate significant differences (α = 0.05) compared with the control (expression at 0 h).

We also performed real-time PCR to obtain insights into expression patterns of TaDGKs under drought stress. TaDGK2A/3A/5B/5A2 were all also induced at 0.5 h, by approximately 2.5- and 4.5-fold, respectively, with the highest expression (8–32-fold increases) observed after 12 h of stress treatment. In contrast to the expression of TaDGKs induced by salt treatment, the transcript level of TaDGK3A, which is higher in leaves, was increased most strongly under drought stress, by a factor of up to 30, while the other TaDGKs were relatively less induced (Fig. 8B). The control gene TaDREB2, which encodes a transcription factors, with major roles in dealing with abiotic stresses, has been demonstrated to be induced under drought stress and salt stress (Djafi et al., 2013; Morran et al., 2011; Carther et al., 2019).

Subcellular localization of TaDGKs

Using the online prediction tool WoLF PSORT, subcellular localization of TaDGK expression was predicted. All cluster I TaDGKs have a trans-membrane region and were predicted to be distributed among multiple cellular organelles, though mainly within the nucleus and chloroplast. The cluster II TaDGKs TaDGK6A/B/D were mainly predicted to be localized to the chloroplast and cytoplasm. All TaDGK2/3/7 members were mainly predicted in the nucleus and cytoplasm (Table S9).

We selected TaDGK2A, TaDGK3A, TaDGK5B and TaDGK5A2 proteins for empirically assessment of their predicted subcellular localization. Accordingly, TaDGKs proteins fused to a N-terminal GFP tag, were expressed in tobacco leaves (Fig. S7). Notably, we found TaDGK2A and TaDGK5B were expressed in the nucleus and cytomembrane, while TaDGK3A and TaDGK5A2 were mostly expressed in the cytomembrane based on confocal microscopy (Fig. 9). These results suggested that there was functional differentiation among the members of the TaDGK family. It appears that in addition to catalyzing the production of signaling substances on the cytomembrane, some TaDGKs can could also enter into the nucleus to regulate genes expression.

Figure 9 The subcellular localization of TaDGK2A, TaDGK3A, TaDGK5B, and TaDGK5A2 in tobacco leaves.

(A–D) The subcellular localization of TaDGK2A-GFP (A), TaDGK3A-GFP (B), TaDGK5B-GFP (C) and TaDGK5A2-GFP (D) driven by CaMV35S promoter. Localization of GFP signals from TaDGK proteins fused with GFP. The nuclei are labeled with DAPI. Bright field, epifluorescence, and merged images of tobacco leaves transfected with constructs are shown expressing different fusion proteins. The nucleus (white) and cytomembrane (red) with an arrow. Scale bars are 50,000 nm in length.

Discussion

The DGK family of genes, involved in the metabolism of PI as catalytic enzymes, play pivotal roles in the production of lipid messengers that mediate plant growth, development, and responses to biotic and abiotic cues (Wang et al., 2014; Kimura, Jennings & Epand, 2016). We identified 24 TaDGK genes in wheat, which contain conserved DGKa, DGKc and two C1 domains. The TaDGKs, encoded proteins ranging from 488–723 aa, were distributed along the 18 of 21 wheat chromosomes (Fig. 1). A phylogenetic analysis, based on the inferred protein sequences of TaDGKs and other species, revealed that the TaDGKs were grouped into three clusters, and more aptly classed with DGKs from the monocots rice and maize than those from the dicots Arabidopsis and soybean. Structures analysis demonstrated that TaDGKs in the same cluster had similar exon-intron structures. For example, the TaDGKs in cluster II and III had 12 exons, while those in cluster I had 7 exons (Figs. S3A, S3B). In addition, we found some genes, such as TaDGK1D, TaDGK6B2, TaDGK7A/B/D and TaDGK7B2/D2, possess a long intron containing transposon elements (Table S6), which implied these transposon elements may regulate the biological function of TaDGKs as potential controlling elements. These data provide fundamental information about TaDGKs, which lays a foundation for the subsequent research on TaDGK function.

DGKs are involved in responses to osmotic stress in plants

Various lipids are related in controlling plant growth, development, and dealing with biotic and abiotic stresses, and their synthesis is modulated by lipid-signaling enzymes (Wang, 2004; Margutti et al., 2017). The PLC/DGK pathway is one of the most important signaling networks in response to biotic and abiotic stresses (Hou, Ufer & Bartels, 2016; Arisz, Testerink & Munnik, 2009; Munnik & Vermeer, 2010). Our previous work revealed the function of TaPLC1 in controlling seedling growth and adapting to drought and salt stress (Zhang et al., 2014), while the present study is focused on the role of DGK genes in wheat.

DGKs are widely distributed in eukaryotes. In silico identification has been used for the functional prediction of DGKs family members in Arabidopsis, rice, maize, apple, and soybean (Arisz, Testerink & Munnik, 2009; Sui et al., 2008; Ge et al., 2012; Li et al., 2015a; Li et al., 2015b; Carther et al., 2019). Research assessing the response of DGKs to osmotic stress has mainly been performed through analysis of transcriptome and examination of mutants. AtDGK1 was the first cloned plant DGK cDNA (Katagiri, Mizoguchi & Shinozaki, 1996). Both AtDGK1 and AtDGK2 genes were induced under low temperature (4 °C) (Gómez-Merino et al., 2004; Lee, Henderson & Zhu, 2005). Later, AtDGK1 and AtDGK2 were determined to be cold-responsive genes using Affymetrix GeneChips (Lee, Henderson & Zhu, 2005). In Arabidopsis, AtDGK2, DGK3, and DGK5 also mediate the response to cold (Gómez-Merino et al., 2005; Gómez-Merino et al., 2004; Tan et al., 2018). In line with this observation, dgk2, dgk3, and dgk5 revealed improved tolerance and decreased PA synthesis under freezing temperatures (Tan et al., 2018). Under optimal and high salinity conditions, double mutants dgk3 dgk7, dgk5 dgk6, and dgk1 dgk2 exhibited lower germination rates, lower total respiration rates, use of an alternative respiratory pathway, and lower PA content in response to 24-epibrassinolide (EBL) treatment compared to wild-type plants (Derevyanchuk et al., 2019). In addition, all three ZmDGKs, six of the eight DGKs in the apple genome, and almost all GmDGKs had significantly induced expression under PEG or salt treatments (Sui et al., 2008; Ge et al., 2012; Carther et al., 2019). Although the functions of plant DGKs still need further in-depth and comprehensive analysis, these findings confirmed the role of DGKs in osmotic stress, and it was helpful to study the relationship between the paired PLC/DGK pathway and environmental stresses in plant responses (Escobar-Sepúlveda et al., 2017).

Expansion of the DGK gene family in hexaploid wheat

In this study, we identified TaDGKs and analyzed their evolution and expression. We isolated 24 TaDGKs in hexaploid wheat (T. aestivum) using a genome-wide approach. The number of DGKs in wheat is approximately three-fold higher than that in Arabidopsis (i.e., seven DGKs) (Gómez-Merino et al., 2005), rice (eight DGKs) (Ge et al., 2012), and other plant species (Sui et al., 2008; Li et al., 2015a; Li et al., 2015b; Carther et al., 2019). This is because the ancestor of allohexaploid bread wheat (T. aestivum) underwent three polyploidizations (Marcussen et al., 2014), and segmental or tandem duplication events of DGK genes led to the expansion of gene families throughout plant genome evolution (Kashkush, Feldman & Levy, 2015). Accordingly, three groups of genes corresponding to TaDGKs in the A, B, and D subgenomes formed clusters with bootstrap values of 1000 (Fig. 4). These events, such as intrachromosomal serial replication and gene-loss, may have occurred during the evolution process, whereas a gene loss event appears to have occurred among TaDGKs on wheat chromosomes 7A, 6A and 6D (Fig. S2).

DGK family members in mammals are grouped into five subtypes (Topham & Prescott, 1999), whereas plants DGKs fall into three distinct clusters (I, II, and III). All plant DGK genes within Cluster I, which consist of the most intricate plant DGKs, most closely similar with the DGK genes in Type III, which are the most basic DGKs in mammalian cells (Arisz, Testerink & Munnik, 2009). Notably, plant phosphoinositide-dependent phospholipases C (PI-PLC) also closely resembles the most basic PLCζ isoform of mammals (Pokotylo et al., 2014). This shows the difference in the DGK/PLC pathway between plant and animal cells, and it is unknown whether this structural simplification indicates functional simplification.

The results of phylogenetic analysis are further supported by the analyses of sequence and structural features of TaDGK genes. TaDGK homologs in the same clade, such as wheat and rice, own similar intron-exon gene structures (Fig. S4). In addition, the structural organization of TaDGKs within clusters I and II comprised seven and twelve exons, respectively, consistent with other plant species, such as rice, apple, and soybean (Li et al., 2015a; Li et al., 2015b; Carther et al., 2019). This conservation of gene structures across species shows the plant DGK family is also conserved in its genomic structure. It should be noted that the TaDGK1D, TaDGK6B2, TaDGK7A/B/D and TaDGK7B2/D2 genes have very long introns. We found that these long introns contained many different types of transposon elements (Table S6), which suggested the formation of them occurred through the insertion of transposons as potential controlling elements (Kashkush, Feldman & Levy, 2015). These transposons may regulate TaDGKs biological functions by blocking their transcription (Fig. S3 and Table S6).

Expression and functional divergence of TaDGK genes

Usually, the gene expression patterns, in various tissues and organs, were detected to analyze corresponding biological functions. Based on the in-silico assessment of RNA-seq experiments and subsequent qRT-PCR confirmation, TaDGKs were observed to exhibit specific expression in the root, shoot, leaf, spike, and grain (Fig. 7), implying that they may play a role in specific growth and development processes. Although the expression level of each TaDGK copy gene among the different chromosomes differed slightly from each other, the trend for each homolog was similar. The results suggest that TaDGK2/5/5-2/6 appear to be involved in root growth and development, while TaDGK1/3, TaDGK7B2/D2 and TaDGK6B2 may be related to grain development. The closest ortholog to TaDGK2 is OsDGK1 in rice, which is also highly expressed in roots and affects rice lateral root development and seminal root/ crown root growth (Yuan et al., 2019). In wheat, the expression of all TaDGKs in leaves was significantly lower than that in roots. Unlike wheat, all MdDGKs in apple showed high expression in stems (Li et al., 2015a; Li et al., 2015b), and almost all GmDGKs in soybean showed noteworthy expression levels in leaves and roots, but no significant contrasts in expression levels between leaf and root tissues (Carther et al., 2019).

Many stress response elements and hormone response elements were found in the promoter regions of TaDGKs, suggesting that TaDGKs play a role in responding various stress and hormone signaling. We examined the expression profile of TaDGKs under osmotic stress and phytohormone treatments using publicly available microarray datasets. All TaDGK1A/B/D and TaDGK2D genes exhibit significantly increased transcript levels at 4 °C compared to 23 °C (Fig. S6 and Table S8). Similarly, AtDGK1, 2, 3, and 5 are induced by exposure to low temperature and contribute to the cold response in Arabidopsis (Gómez-Merino et al., 2004; Carther et al., 2019; Wang, 2004). Low, non-freezing temperatures were found to trigger a very rapid PA increase and were primarily generated through DGK in Arabidopsis suggesting that cold-induced membrane rigidification was upstream of DGK pathway activation (Arisz et al., 2013; Vaultier et al., 2008). Cold treatment caused an increase in the expression of ZmDGK2 and 3 in roots and leaves (Sui et al., 2008). TaDGK1 and AtDGK1/2 belong to cluster I, while TaDGK2D is the closest ortholog to maize ZmDGK2 and 3 in our phylogenetic analysis (Fig. 4). This suggests that sequence homology is related to functional similarity.

ABA may be the most significant factor affecting TaDGKs expression (Fig. S6 and Table S8). Physiological analysis showed that PA triggers early signal transduction events that lead to responses to abscisic acid (ABA) during seed germination. In this reaction, it is the lipid phosphate phosphatase (LPP) AtLPP2, which catalyzes the conversion of PA to diacylglycerol (DAG), is a negative regulator of ABA signaling (Katagiri et al., 2005). Recent research has shown that ABA can stimulate DGK activity independently of AtLPP2 activity in Arabidopsis (Paradis et al., 2011). In short, DGK is involved in ABA signaling but the regulatory mechanism is unclear.

Furthermore, we examined the transcript levels of TaDGK genes under salt and drought stress treatments. According to RNA-seq data in public databases, TaDGK2 and 3 were up-regulated under drought conditions, especially TaDGK3, which was strongly induced (Fig. S6 and Table S8). Our results confirmed that TaDGK3 was most strongly induced by drought. In addition, the transcriptional expression of almost all the tested genes was induced by drought and salt (Fig. 8). Although the expression of all TaDGKs in leaves was significantly lower in wheat, the variation in DGK expression patterns in response to drought or salt stress reflects its metabolic activities in leaves. In addition, it’s an interesting phenomenon that the expression of TaDGKs rapidly increased and then decreased within 24 h under drought and salt stresses. A similar pattern has been found in GmDGKs under abiotic stress (Carther et al., 2019). On the other hand, TaPLCs, another family of key enzymes involved in the inositol phospholipid signaling system, also showed a stress-responsive expression pattern in response to stress (Zhang et al., 2014). We propose that both DGA and PA are upstream signals in the signaling network and that the rapid decrease in TaDGKs expression levels after 12 h, may be a mechanism though which these signals are turn off to prevent their over-amplification of them. Another possibility is that their decrease was caused by damage to the plant becoming too severe.

Conclusion

A total of 24 TaDGK genes were identified from the wheat genome. Based on comparative analyses, we identified putative TaDGKs and inferred their phylogenetic relationships, sequence characteristics, cis-regulatory promoter elements, and subcellular localization patterns. From these results, we obtained insights into the putative functions of TaDGKs, which has the potential to contribute to their further functional dissection in future research. Expression profiles of TaDGKs at a transcriptional level showed each member of this family had specific spatial and developmental expression patterns. In addition, our results indicated that some TaDGKs were significantly induced by salinity or drought stress, suggesting their possible function in responses to environmental stimuli. This will enable the use of TaDGKs for wheat breeding to improve the resistance of wheat to various abiotic stresses. Thus, further research on the biological function of TaDGKs would be beneficial to the development of drought- and salt-resistant of wheat.

Supplemental Information

Supplemental Information 1 Gene ontology analysis (biological process and molecular function) of TaDGKs

Click here for additional data file.

Supplemental Information 2 Phylogenetic analyses on TaDGK6B2, TaDGK7B2/D2 and their orthologs from genomes of ancestral species

The numbers beside the branches indicate the bootstrapvalues that support the adjacent nodes.

Click here for additional data file.

Supplemental Information 3 Phylogenetic relationship, gene structure, and motifs of TaDGKs

(A) Phylogenetic analysis, with different colors indicating genes in individual clusters. (B) Schematic diagram of exon/intron structures of DGK genes in rice and wheat. Red, purple, and green boxes represent upstream regions, CDSs, and downstream regions, respectively. (C) Schematic diagram of the DGK gene motifs. Different colored boxes are assigned to each motif.

Click here for additional data file.

Supplemental Information 4 Phylogenetic tree and gene structure of DGK family members in rice and wheat

(A) Phylogenetic analysis, with different colors indicating genes in individual clusters. (B) Schematic diagram for exon/intron structures of DGK genes in rice and wheat. Red, purple, and green boxes represent upstream regions, CDSs, and downstream regions, respectively.

Click here for additional data file.

Supplemental Information 5 Detailed information on motifs

Click here for additional data file.

Supplemental Information 6 Expression of TaDGK genes under different treatment

Click here for additional data file.

Supplemental Information 7 Subcellular localization vector construction

Subcellular localization vector construction for (a) 35S:TaDGK2A-GFP, (b) 35S:TaDGK3A-GFP, (c) 35S:TaDGK5B-GFP, and (d) 35S:TaDGK5A2-GFP. (A) Amplification of TaDGK2A CDS and TaDGK3A CDSs. (B) Amplification of TaDGK5B CDS and TaDGK5A2 CDS. (C) Colony PCR verification of pEASY-TaDGK2A and pEASY-TaDGK3A constructs. (D) Colony PCR verification of pEASY-TaDGK5B and pEASY-TaDGK5A2 contructs. (E). Double enzyme SpeI/KpnI digest for pEASY- TaDGK2A; Double enzyme XbaI/KpnI digest for pEASY-TaDGK3A. (F). Double enzyme SalI/SpeI digest for pEASY-TaDGK5B; double enzyme Xba I/Kpn I digest for pEASY-TaDGK5A2. (G). Double enzyme SpeI/KpnI digest for 35S:TaDGK2A-GFP; double enzyme Xba I/Kpn I digest for 35S:TaDGK3A-GFP. (H). Double enzyme SalI/SpeI digest for 35S:TaDGK5B-GFP; double enzyme XbaI/KpnI digest for 35S:TaDGK5A2-GFP.

Click here for additional data file.

Supplemental Information 8 GO annotation analysis of TaDGK genes

Click here for additional data file.

Supplemental Information 9 Primers used for subcellular localization and quantitative real-time PCR analyses

Click here for additional data file.

Supplemental Information 10 Nucleic acid, deduced amino acid, promoter, and genomic sequences of TaDGK genes

Click here for additional data file.

Supplemental Information 11 Properties of DGKs identified from the wheat genome

Characterization of the wheat (Triticum aestivum L.) genome data transcript identifiers (IDs) with gene names, gene loci, gene identifiers, chromosome (Chr), localization coordinates, ORF length (bp), isoelectric point (pI), protein identifiers, protein lengths (amino acids; aa), and protein weights (kDa).

Click here for additional data file.

Supplemental Information 12 GenBank accession numbers of DGKs used for Phylogenetic analyses

Click here for additional data file.

Supplemental Information 13 Transposable element analyses on the intron of TaDGKs

Click here for additional data file.

Supplemental Information 14 Putative regulatory cis-elements in TaDGK gene promoters

Click here for additional data file.

Supplemental Information 15 Expression of TaDGK genes under hormone treatment and abiotic stress

Click here for additional data file.

Supplemental Information 16 Subcellular location and trans-membrane region of TaDGKs

Click here for additional data file.

Supplemental Information 17 Raw Data of Figs. 7B, 7C and 8

Click here for additional data file.

The authors would like to acknowledge Yan Li and Zhuo Chen for providing Chinese Spring wheat seeds and the pSUPER1300 vector.

Abbreviations

DGK diacylglycerol kinase

DAG diacylglycerol

PA phosphatidic acid

PI phosphatidylinositol

PI-cycle phosphatidylinositol cycle

PLC phospholipases C

ORF open reading frame

aa amino acid

DREB dehydration-responsive element-binding protein 2

GFP green fluorescent protein

qRT-PCR quantitative real-time PCR

Additional Information and Declarations

Competing Interests

Author Contributions

DNA Deposition

Data Availability

The authors declare there are no competing interests.

Xiaowei Jia performed the experiments, authored or reviewed drafts of the paper, and approved the final draft.

Xuyang Si performed the experiments, analyzed the data, prepared figures and/or tables, authored or reviewed drafts of the paper, and approved the final draft.

Yangyang Jia performed the experiments, prepared figures and/or tables, and approved the final draft.

Hongyan Zhang and Wenjing Li analyzed the data, prepared figures and/or tables, and approved the final draft.

Shijun Tian performed the experiments, analyzed the data, prepared figures and/or tables, and approved the final draft.

Ke Zhang and Yanyun Pan conceived and designed the experiments, authored or reviewed drafts of the paper, and approved the final draft.

The following information was supplied regarding the deposition of DNA sequences:

The sequences are available in GenBank: TaDGK1A (KAF6984428.1), TaDGK1B (KAF6989925.1), TaDGK1D (KAF6995613.1), TaDGK2A (KAF7004576.1), TaDGK2B (KAF7011950.1), TaDGK2D (KAF7019302.1), TaDGK3A (KAF7024027.1), TaDGK3B (KAF7031092.1), TaDGK3D (KAF7038111.1), TaDGK5A (KAF7057220.1), TaDGK5B (KAF7063829.1), TaDGK5D (KAF7092610.1), TaDGK5A2 (KAF7057607.1), TaDGK5B2 (KAF7064212.1), TaDGK5D2 (KAF7071278.1), TaDGK6A (KAF7080894.1), TaDGK6B (KAF7086390.1), TaDGK6B2 (KAF7086908.1), TaDGK6D (KAF7091572.1), TaDGK7A (KAF7095996.1), TaDGK7B (KAF7101802.1), TaDGK7B2 (KAF7103042.1), TaDGK7D (KAF7109040.1), TaDGK7D2 (KAF7110201.1).

The following information was supplied regarding data availability:

The raw measurements are available in the Supplemental FIles.

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
