# Peer review of "Genomic profiling and expression analysis of the diacylglycerol kinase gene family in heterologous hexaploid wheat"

_PeerJ, doi:10.7717/peerj.12480_

## Round 0.1 · original submission · Major Revisions

Please address the reviewers' comments and either modify the manuscript or answer their comments.

Reviewer 1 ·

Basic reporting

Authors have used clear and professional English throughout the manuscript.

Experimental design

The experimental design is standard.

Validity of the findings

The study is comprehensive and provides all underlying data.

Additional comments

Dear Authors,
Congratulations for this interesting study. The manuscript by Jia et al. reported the genomic profiling and expression analysis of the 20 DGK gene family members from the wheat genome. The manuscript is well written. The results and conclusions of the papers are interesting and convincing. However, I found some concerns that need to be addressed before considering this manuscript for publication.

1. I am wondering about the purpose of this study. Most of the previous studies with different plants reported the similar results. The authors could have discussed why this study is important and how these findings added significantly to the already established facts about the DGK gene family.
2. The introduction part is well written but I think the authors should include more details about the DGK family isoforms, classes and structural insights. The authors should add – how the stresses involve important signaling molecules related to diacylglycerol metabolism.
3. Methods have been described with sufficient details but I think in the “Plant transformation” section clarity is missing. Authors should add more details to it.
4. The authors have primarily used the BLAST program to identify the TaDGK genes. How did the authors make sure that they are not missing any TaDGK genes which are distantly matching to Arabidopsis and rice gene queries?
5. What criteria the authors have followed to shortlist the four genes for expression analysis? Why were no genes from cluster III considered?
6. In the phylogenetic analysis the DGK genes from plants like Malus prunifolia, Brassica rapa, Cicer arietinum, Coffea canephora, Solanum tuberosum, Sorghum bicolor etc were missing. Any explanation for these exclusions.
7. The Figure 4 (multiple alignments) should be placed prior to the figures related to phylogenetic analysis (figure 2/3) in the text.
8. Kindly move figure 2 or figure 6 to the supplementary figure section. They are basically depicting the similar results. And it is hold true in between figure 1 and table 1.
9. The genes used for RNA-seq analysis (figure 8A) must be used for tissue specific expression analysis (figure 8B).
10. The qRT-PCR results of TaDGK genes under salt stress and drought conditions (figure 9) showed that gene expressions level after 24 h were lower than 12 h. What is the possible explanation? Why is gene expression after 12 h is maximum for all the genes studied, why is it dropping when the authors increase the treatment time? Please discuss this down-regulation in the discussion section. I could not find the expression level after 48 h of treatment in the figure but it is mentioned in the figure legend.
11. Please provide the NCBI accession number of all the 20 TaDGK genes.
12. Why were only two genes from the same cluster used for subcellular localization study? At least one gene from each cluster should be used.
13. The “Result and Discussion” section should be improved. The authors should discuss the findings instead of only reporting the outcome of the experiments.
14. For the salinity and drought treatments, 200 mM NaCl and 20% PEG6000 was used respectively. How were these concentrations fixed at these levels? Did the authors perform concentration gradient experiments? The citations should be mentioned if the authors followed any previous reports.
15. Please discuss about the wheat BT009326 gene. Is it related to any of the 20 genes discussed here?

Minor Comments:
1. Line 34- Change the parentheses style for phosphatidylinositol 4,5-bisphosphate. It should be [PI (4, 5) P2].
2. Line 54 and 55 need a citation.
3. Line 71- Please provide the web address of BLASTN.
4. Line 88- Please provide the web address of Evolview tool.
5. Line 117- Please number the tables according to their mention in the text e.g. here S5 should be S1.
6. Line 119- Please italicize “Nicotiana benthamiana”.
7. Line121- Please correct the enzyme writing style e.g. no space between “Spe” and “I”.
8. Line 127- Please italicize “Agrobacterium tumefaciens”.
9. Figure legend of figure 3- The number instead of “he number”

Reviewer 2 ·

Basic reporting

No comment.

Experimental design

No comment.

Validity of the findings

No comment.

Additional comments

This manuscript described by Jia et al. performed genomic profiling and expression analysis of the diacylglycerol kinase (DGK) gene family in wheat. They identified 20 TaDGKs and found that several TaDGKs were strongly induced by salt and drought stresses. Because many DGK genes have been already identified in several plants including Arabidopsis thaliana, rice, tobacco and soybean, the findings in this manuscript are substantially novel. However, their findings will be added to the database of DGK in plants and provide some insights into the putative functions of plant DGKs.

There are several concerns that should be addressed by the authors (generally lack of explanation).
(1) Line 177–185: Are exon/intron boundaries of catalytic domains conserved among all TaDGKs?
(2) Line 192: What are the motifs? Are the motifs related to domains of DGK proteins (e.g., catalytic and C1 domains)?
(3) Line 209–224: There are some contradictions between microarray-based expression pattens and those obtained by real-time PCR (e.g., TaDGK3A: roots (microarray) vs. stems (real-time PCR). The authors should discuss.
(4) Line 229: 10 min is not correct. This would be 6 min (0.1 h in Fig. 9).
(5) Line 231 and 236: The authors mentioned the results (increases) after 12 h (Fig. 9A and B). However, the values were decreased after 24 h. Why? The authors should discuss.
(6) Line 239–240: The authors compared expression pattens of TaDGKs under abiotic stresses with only TaDREB2. In addition, the authors should compare with DGKs in other plants.
(7) Fig. 3: What is the pink domain?
(8) Fig. 10: What is DIC? Where are the nucleus and cytoplasm? The authors should indicate them with arrows.

Reviewer 3 ·

Basic reporting

- The article provides sufficient literature, but adding suggestions regarding the application of the findings would be helpful. It is important to identify the significance of these findings/work.

- Figure 1: Chr7D and TaDGK7D are missing in the figure
- Figure 2: What do the values mentioned in the tree branches signify?
- Figure 3 legend: Line 2 - "T" is missing in the first word
- Figure 5: Labels for each colored box (blue, light green, dark green, black) can be listed
- Figure 6A and 6B are the same as supplementary fig 1 (fig S1) and therefore redundant.
- Figure 8B - the purple color box is denoted for "Ears" but in the figure legend its mentioned as "spikes"
- Figure 9: Since the qPCR values are compared to 0hr data (it's understood by default that the relative expression level at 0hr is 1), authors can remove the 0hr time point data from the graph.
-Figure 9: Figure legend mentions the 48hr timepoint but the data is absent from the bar graph
- Figure 10: scale bar is missing, labels should be outside the images
- Figure 10: figure legend mentions the brightfield image but the images are labeled as DIC. The author needs to correct this.

Experimental design

Methods section
2.4 RNA seq data: (line 97-99) Authors need to cite appropriate references and the study IDs (if any) for the three datasets used.

2.6 RNA extraction: Line 112-113. The first line mentions that total RNA was extracted, but in the second line gDNA is mentioned. The authors need to correct this method section.
Also, details on methods used to calculate the relative expression levels need to be described (2^-ΔΔCT method?)

2.7 plant transformation: Authors need to cite the reference, as the protocol for transformation is not described in detail.

Validity of the findings

Authors need to provide details in the method section regarding the statistical test/s applied for the qPCR data.

Additional comments

Authors identified 20 diacylglycerol kinase genes, an important enzyme in the Phosphatidylinositol cycle in the hexaploid wheat. The authors performed a detailed bioinformatic and expression analysis of the genes and showed that they were significantly induced under environmental stress of salinity and drought. This is an interesting work that is well performed with all the raw data organized in the supplementary files.

Major comments:
1. Identification, chromosomal location, and gene:
- Line 77: some DGK genes lacking catalytic domain are ignored - authors should list the number and name of genes ignored. Can authors also comment on why the following genes were not included in the analysis
TraesCS7A02G297200 - Chr7A
TraesCS7B02G198600 - Chr7B
TraesCS7D02G291600 - Chr7D
TraesCS6B02G428500 - Chr6B

- Did authors name the genes (TaDGK1-6A/B/D, TaDGK7B/D), or were they already named? It looks like the naming system is based on chromosome number. It is confusing since TaDGK4A/B/D and TaDGK5A/B/D are all on chr 5 and not on Chr 4. If the authors have named the genes, the naming should be corrected (i.e for eg. TaDGK5A1 and TaDGK5A2)

2. Subcellular localization
- The authors mention that TaDGK2A and 3A are localized in the cytoplasm. But looking at the images, it seems the protein is present in the cell periphery (cell membrane). The authors should explain this.
- TaDGK2A in addition to its localization in the cytoplasm, is also localized in the nucleus. It would be good to confirm the nucleus localization with DAPI and its colocalization with TaDGK2A-GFP.

Minor comments:
Line 13: Full form of PI
Line 22: "express" word can be deleted
Line 36: "These derivatives and catalytic enzymes play....." - Authors should mention which catalytic enzymes?
Line 107: Timepoint 1/2 or 0.5hr is missing. Timepoints for salinity and drought stress needs to be corrected throughout the paper and supplementary files. At some places it is mentioned as 0.1hr,0.5hr, etc., and at others, as 10min, 30 mins. Just make sure 10 min is not the same as 0.1hr.
Line 135-136: Is it 14 chromosomes or 17 chromosomes
Line 169 and figure 4A: The authors mention ATP binding site with a GXGXXG consensus sequence (red box of fig 4A). But the red box in the figure does not match the GXGXXG consensus. This needs to be corrected.
Line 217-240: Why were only genes TaDGK2/3/4/5 selected for qPCR and not TaDGK1/3/7?
Line 254: what do authors mean by "its close relatives"?
Line 254-261: As stated earlier, it is important to add 1-2 lines on the application or use of this data and/or future work.

---

## Round 0.2 · Minor Revisions

Two reviewers are satisfied with the revision while one reviewer has some minor comments.

Reviewer 1 ·

Basic reporting

The authors have used clear and professional English throughout the "revised manuscript". Authors have provided proper literature references.

Experimental design

No comments

Validity of the findings

No comments

Additional comments

The authors have satisfactorily addressed all of the concerns raised by me. The "revised manuscript" is now in acceptable form. Congratulations to the authors for this nice Job.

Reviewer 2 ·

Basic reporting

no comment

Experimental design

no comment

Validity of the findings

no comment

Additional comments

The authors have modified the manuscript accordingly to the suggestions proposed, and this has improved in clarity. I have no more issues to address.

Reviewer 4 ·

Basic reporting

Line 126: Please change ‘ware’ to ‘were’.
Line 138: Change ‘cdna’ to ‘cDNA’.
Line 170: BLAST cannot be used as a verb. Please modify the sentence that uses ‘blasted’.
Line 176: Change ‘was lost’ to ‘were lost’. Change ‘befor’ to ‘before’.
Line 193: Remove repeated phrase ‘as in other studied plants’.
Line 237: Expand IBS, either here or at first usage.
Figure 1, Please replace ‘Clustel’ with ‘Cluster’ in the color key at the bottom of the figure.
Lines 183-184: Move this sentence to the end of the paragraph. Otherwise it seems that the following 2-3 sentences are about BT009326 and TaDGK2B.
Line 634, Figure S2 legend: I believe Fig. S2B is only wheat DGKs. The legend says wheat and rice.
Fig. S3: (A) and (B) labels are missing.
Fig.6: Main figure should stand alone without having the need to refer to supplementary material. The meaning of each capital letter should be provided here in Fig. 6 itself in a small table.
Line 266: Table S7 is cited here while it does not show the data explained here.

Experimental design

Line 144: 2ΔCt should be 2^-ΔCt and 2-ΔΔCt should be 2^-ΔΔCt

Validity of the findings

Lines 259-261: What could be the reason for ABA-mediated repression of TaDGK2,3, and 6 when there is no ABA-sensitive motif (A) found in the upstream region of these genes?

Additional comments

The manuscript should be edited again for any grammar/spelling mistakes. I have suggested a few changes, but there could be more.

---

## Round 0.3 · Minor Revisions

Dear Dr. Zhang,

Apologies for the delay. Your paper is almost ready to be accepted for publication in PeerJ.

Before we can do that, please address these comments from the Section Editor:

> The manuscript reads well; however, there are issues with the version of the reference genome used from the Ensembl source (there are multiple version presentations). There appears to be some refinement of the DGK roles; however, to provide better identification would require the version and coordinate values (likewise poor description of progenitor species sources and versions). Likewise, as functional roles are being defined it would be important to introduce annotation values such as that associated with gene ontology, none is stated.

> Journal manuscripts are often scanned by text-mining software that locates and extracts core data elements, like gene function. Adding standard ontology terms, such as the Gene Ontology (GO, geneontology.org) or others from the OBO foundry (obofoundry.org) can enhance the recognition of your contribution and description. This will also make human curation of literature easier and more accurate. None of this was visible.

> Does the Triticeae Multiomics Center have an established domain name as it is only presented as an IP address that may not be available in the future through a literature reference, or established resource? If this is the case relevant data should be deposited in an established third-party data collection resource. It is also confusing when there are Materials and Methods provided for data that is stated as being downloaded from a resource; what was actually done on the bench apart from that downloaded and analyzed.

> The manuscript in general is an interesting read but lacks presentation in an organized fashion for the reader to connect to the 24 DGK genes being discussed. The manuscript is in need of further revision.

See also the following specific edits:

EDITS
LINE NO: / BEFORE / AFTER / [COMMENTS]
LINE 97: / BlastP / BLASTP / [consistency in style of presentation.

---

## Round 0.4 · Minor Revisions

Please address the following Section Editors comments and corrections:

> The manuscript does seem a bit improved, but is still unclear in many areas. There also appears to have data mentioned which is not presented, and without clarifications. There were also a few notable edits suggested as noted below. As data is missing I would rank this as a major revision until that gap can be filled.

> Why are only 17 chromosomes mentioned when there are 18 displayed in the figure; it is confusing when only the matched chromosomes are mentioned and not the entire set as reference, yet there is no mention of the normal number of chromosomes expected in wheat and discussed why none were found in the remaining chromosomes analyzed. I did not see any partitioning of data with regards to GO annotation though it is mentioned in the manuscript; it did not appear in the manuscript nor the supplemental files.

> In table S7 there are values presented; however, if adding annotations it would be appropriate to add annotation terms regarding tissue. The result are arbitrary without committing toward an actual assessed value. It seems that raw data is merely being presented without coming to any conclusions.

> Journal manuscripts are often scanned by text-mining software that locates and extracts core data elements, like gene function. Adding standard ontology terms, such as the Gene Ontology (GO, geneontology.org) or others from the OBO foundry (obofoundry.org) can enhance the recognition of your contribution and description. This will also make human curation of literature easier and more accurate. None of this was visible.

EDITS
LINE NO: / BEFORE / AFTER / [COMMENTS]
LINE 20: / Chinese spring / Chinese Spring / [.]
LINE 86: / genome-wide of wheat. / genome-wide studies of wheat. / [.]
LINE 110: / . / . / [Gene ontology (GO) enrichment. WHERE IS THE DATA?]
LINE 132: / and then that were / and then they were / [.]
LINE 149: / independent samplest test. / ? / [ Do you mean t—test? ]
LINE 166: / detection 461 nm / detection at 461 nm / [.]
LINE 172: / domain searchers were / domain searches were / [.]
LINE 175: / along the 17 wheat chromosomes / along the 18 0f 21 wheat chromosomes / [ very confusing? ]
LINE 189: / wheat chromosome / wheat chromosomes / [.]
LINE 216: / more aptly classed with DGKs / more aptly classified with DGKs / [.]
LINE 297: / four member of TaDGK family / four members of the TaDGK family / [.]
LINE 333: / the 17 wheat chromosomes / the 18 of 21 wheat chromosomes / [ confusing? ]
LINE 365 : / unthorough, / ? / [ what is being said here! Confusing! ]
LINE 365 : / was benefit of researching / . / [ what is being said here! Confusing! ]
LINE 378: / wheat chromosome / wheat chromosomes / [.]
LINE 384: / PLCζ / . / [ Is a special character used here correct? ]
LINE 388: / own semblable exon-intron structures / . / [ semblable? What is this? ] "

Reviewer 1 ·

Basic reporting

NA

Experimental design

NA

Validity of the findings

NA

Additional comments

I carefully read the revised version of this manuscript. In the revised version the authors have carefully introduced the changes suggested by the section editor. I found this manuscript can be now published.

---

## Round 0.5 · accepted · Accept

The authors have addressed the editor's comment satisfactorily.